# Online Handwriting Recognition Method with a Non-Inertial Reference Frame Based on the Measurement of Linear Accelerations and Differential Geometry: An Alternative to Quaternions

**DOI:** 10.3390/mi15081053

**Published:** 2024-08-21

**Authors:** Griselda Stephany Abarca Jiménez, Carmen Caritina Muñoz Garnica, Mario Alfredo Reyes Barranca, Jesús Mares Carreño, Manuel Vladimir Vega Blanco, Francisco Gutiérrez Galicia

**Affiliations:** 1Unidad Profesional Interdisciplinaria de Ingeniería Campus Hidalgo (UPIIH), Instituto Politécnico Nacional (IPN), Carretera Pachuca—Actopan Kilómetro 1+500, Distrito de Educación, Salud, Ciencia, Tecnología e Innovación, San Agustín Tlaxiaca 42162, Hidalgo, Mexico; gabarcaj@ipn.mx (G.S.A.J.); mvvegab@ipn.mx (M.V.V.B.); fgutierrezga@ipn.mx (F.G.G.); 2Electrical Engineering Department, CINVESTAV IPN, Av. Instituto Politécnico Nacional 2508, Gustavo A. Madero, San Pedro Zacatenco, Mexico City 07360, Mexico; cmgarnica@cinvestav.mx (C.C.M.G.); mreyes@cinvestav.mx (M.A.R.B.)

**Keywords:** handwriting, non-inertial frame of reference, linear acceleration, inertial measurements, differential geometry, feature engineering, quaternion

## Abstract

This work describes a mathematical model for handwriting devices without a specific reference surface (SRS). The research was carried out on two hypotheses: the first considers possible circular segments that could be made during execution for the reconstruction of the trace, and the second is the combination of lines and circles. The proposed system has no flat reference surface, since the sensor is inside the pencil that describes the trace, not on the surface as in tablets or cell phones. An inertial sensor was used for the measurements, in this case, a commercial Micro-Electro Mechanical sensor of linear acceleration. The tracking device is an IMU sensor and a processing card that allows inertial measurements of the pen during on-the-fly tracing. It is essential to highlight that the system has a non-inertial reference frame. Comparing the two proposed models shows that it is possible to construct shapes from curved lines and that the patterns obtained are similar to what is recognized; this method provides an alternative to quaternion calculus for poorly specified orientation problems.

## 1. Introduction

Human communication modes have evolved from drawings to patterns such as letters or symbols; we have seen the representation of these patterns in stone, paper, and nowadays in digital systems. One of the most common ways of communication is through handwriting, which is nothing more than a set of letters or patterns that together encode a message, making it possible for people who know this code to receive and recognize it. Currently, with the growing use of information technologies, there is a need to digitize messages to make them reach a broader number of people, and it is also necessary to have a way to analyze what is digitized. Two essential concepts appear when discussing this handwriting analysis: online (recognition in real time) and offline (recognition in non-real time). There are various efforts to improve the online handwriting systems to reproduce faithfully the strokes of the pen movement [1,2,3,4,5,6,7,8], and these models are based on known and efficient data analysis methodologies. However, they require prior knowledge of the patterns; that is, they need to know in advance what the patterns should be like in order to reconstruct them, and in some other cases, it is necessary to have the trajectory of the stroke to perfect it. The works [1,2,3,4,5,6,7,8] use deep learning algorithms that insert the electrical characteristics of the sensors; in other cases, the focus is on optimizing the traces already made on a surface.

Although it is a simple task, handwriting is a highly complicated skill. It is necessary to have a mental picture or memory of what each letter looks like and then develop a motion plan to create it as it is stored in our memory. The development of higher-level handwriting skills (letter formation, sizing, spacing, alignment, and more) all begins with being able to copy simple shapes (prewriting shapes) [9,10,11].

From these prewriting shapes (i.e., horizontal lines, circles, vertical lines), it is time to move on to the letter drawing (how the letters are represented) and construction (how each letter is produced). Efficient construction typically leads to better legibility and speed. Widely known online writing systems have a fixed reference system, either through an artificial vision sensor, which establishes the usual order of a matrix in rows and columns and correlates them with the position in the viewing angle vision of the person who writes [12], or through a finite-dimensional tablet and a pen, establishing fixed references with an orthogonal Cartesian coordinate frame according to the tablet. This work intends to generate a trace as faithful as possible to a path made by the user without depending on a fixed surface or being in a specific vision area. The main idea is to establish a digital writing system with two-dimensional strokes that do not require a support surface; this is writing on the fly.

It is worth mentioning that the goal of the methods proposed in this work is online handwriting recognition without a fixed inertial frame of reference, like a screen, tablet, or any other reference surface. Given the above, we do not count on a fixed reference to establish a coordinate system, so the proposed methods must not rely on a fixed reference frame; this is the main differentiator against classic online handwriting recognition methods and devices, and quaternions are a popular and efficient alternative to solving this problem and creating orientation systems using IMUs [13,14,15,16,17,18,19,20,21,22]. However, their use implies a more complex mathematical model than the one presented here.

The present work focuses on generating patterns with defined shapes from geometric characteristics, representing the information of the letters in two formats: the points in Cartesian coordinates to graph the strokes directly or the polar coordinates of the stroke that allow the generating of normalized vector features instantly. Furthermore, the methods presented here do not require prior knowledge of the patterns; they trace them on the fly. Finally, this work seeks to generate high-quality patterns to guarantee better performance in future processing using an artificial intelligence system, as mentioned in [23,24,25]. The quality of a prediction system’s results is largely based on the quality of the input data.

The system developed for the experimentation comprises three parts: the perception system, consisting of a three-axis IMU (Inertial Measurement Unit), a data acquisition and processing board, and a delivery system for the trace display.

## 2. Materials and Methods

The hardware of the digital writing system comprises an IMU, a data acquisition and processing system, a 3D printed case and a workstation. A pictorial representation of the inertial system and the IMU’s inner axis are shown in Figure 1; the IMU used is the MPU6050 module, of which the 3-axis accelerometer and the 3-axis gyroscope are used, and an Arduino UNO board is used for data acquisition, information processing, and communication to the workstation, the latter being the one that allows viewing the reconstructed trace according to the proposed mathematical models (see Figure 2).

A simple user-friendly case for the prototype was made with a 3D printer, with the components placed inside and aligned in the desired position (see Figure 1) for the proper operation of the device, as shown in Figure 3. The IMU is fixed to the casing so the user can draw freely, always maintaining the sensing plane between the pen’s tip and the sensor. In this way, although the velocity and acceleration vectors are three-dimensional, only the axes of the sensor plane perpendicular to the pen are sensed.

An algorithm is proposed to reproduce cursive handwriting strokes, using only the IMU data and without a fixed reference frame. In the cursive (often referenced as italic) handwriting style, letters are mainly constructed based on curved shapes, unlike the block letter style. Cursive handwriting has a greater dominance of curvatures, and several characters are usually embedded in a single continuous line. The hypothesis is that any cursive trace can be reproduced by linking a multiplicity of circular sections one after another, by knowing the variation of the slope concerning the arc the trace describes between any given pair of consecutive points$\;P_{i}$ and $P_{i+1}$ (see Figure 4), that is, its curvature K (Equation (1)).

For the practical calculation of the curvature K an angle τ, defined between the tangents to the trace at P and $P^{\prime }$, it is calculated first by means of two consecutive data points obtained from the IMU, then the arc length S between $P_{i}$ and $P_{i+1}$ is calculated from the same consecutive data points, and then the quotient between the angle τ and the arc length S gives the curvature of the piece of trace between $P_{i}$ and $P_{i+1}$.
(1)$$K=\underset{∆S\to 0}{\lim}\frac{∆\tau }{∆S}$$
where
K:Curvature∆τ:Rotation angle$∆S:\mathrm{Arc}\;\mathrm{length}\;\mathrm{between}\;P\;\mathrm{and}\;P^{\prime }$

For the reconstruction of the trace, once the curvature between $P_{i}$ and $P_{i+1}$ is known, a curvature radius is calculated, along with its corresponding center. As the hypothesis is that the trace can be reconstructed by circular sections, once the radius and center of curvature are known, the angular section of the corresponding circle between the two points P and $P^{\prime }$ is calculated based on the arc length previously calculated. All the data are acquired and processed in the Arduino UNO board, and then send to the workstation.

Finally, with all the above data, the circular sections a are plotted through any suitable software in the workstation.

Two different approaches to the proposed algorithm were developed and implemented, called method one and method two. For method one the velocity of the trace is used as the input data, as the selected IMU gives out a rotational velocity. First, this velocity must be transformed to a linear velocity, and for this transformation a turning radius equal to the distance from the tip of the pen to the center of the IMU is assumed. On the other hand, method two uses the linear accelerations directly from the IMU, unlike method one, without any transformation, which reduces the error in the trace reconstruction as it eliminates any assumption regarding any parameter used in the calculations, in addition to allowing the reconstruction of straight lines.

Method one is based only on circular sections, and it works in the way described below. First, suppose that a reconstructed trace is known from the respective in-plane trace velocities; considering the curvature measurement at point P of a curve (Equation (2)), we have
(2)$$k=\left\|\frac{d\vec{T}}{dS}\right\|$$
where
$\vec{T}:\mathrm{Tangent}\;\mathrm{vector}\;\mathrm{to}\;\mathrm{the}\;\mathrm{curve}\;\mathrm{at}\;a\;\mathrm{point}$k:Curvature in absolute valueS:Arc length

Then, under the hypothesis that the trace can be considered as a multiplicity of linked circular sections, the mean arc curvature (the curvature in the section between $P_{i}$ and $P_{i+1}$) can be expressed as in Equation (3), where the change in the tangent vector (Equation (4)) to the curve to be reconstructed is the difference in linear speeds (${\vec{v}}_{i+1}-{\vec{v}}_{i}$) between the points $P_{i}$ and $P_{i+1}$ of the trace; this is assumed to be true, since the tangent vector and linear velocity vector are analogous, as long as the linear velocity component which is tangent to the trajectory is considered. Also, the trace is assumed to be in a plane, so the velocity of the trace can be decomposed into two x and y components (Equation (7)).
(3)$$k_{m}≅\;\frac{\left\|∆\hat{T}\right\|}{∆S}$$
where
$k_{m}:\mathrm{Mean}\;\mathrm{curvature}$$∆\hat{T}:\mathrm{Change}\;\mathrm{in}\;\mathrm{unit}\;\mathrm{tangent}\;\mathrm{vector}$∆S:Change in arc length
(4)$$∆\hat{T}={\hat{T}}_{i+1}-{\hat{T}}_{i}=\frac{{\vec{v}}_{i+1}}{\left\|{\vec{v}}_{i+1}\right\|}-\frac{{\vec{v}}_{i}}{\left\|{\vec{v}}_{i}\right\|}$$
where
${\hat{T}}_{i+1}:\mathrm{Unit}\mathrm{tangent}\;\mathrm{vector}\;\mathrm{at}\;\mathrm{next}\;\mathrm{position}$${\hat{T}}_{i}:\mathrm{Unit}\mathrm{tangent}\;\mathrm{vector}\;\mathrm{at}\;\mathrm{current}\;\mathrm{position}$${\vec{v}}_{i+1}:\mathrm{Velocity}\;\mathrm{vector}\;\mathrm{at}\;\mathrm{next}\;\mathrm{position}$${\vec{v}}_{i}:\mathrm{Velocity}\;\mathrm{vector}\;\mathrm{at}\;\mathrm{current}\;\mathrm{position}$
(5)$$\left\|∆\hat{T}\right\|=\frac{\sqrt{{\left(\left\|{\vec{v}}_{i}\right\|v_{{i+1}_{x}}-\left\|{\vec{v}}_{i+1}\right\|v_{i_{x}}\right)}^{2}+{\left(v_{{i+1}_{y}}\left\|{\vec{v}}_{i}\right\|-\left\|{\vec{v}}_{i+1}\right\|v_{i_{y}}\right)}^{2}}}{\left\|{\vec{v}}_{i}\right\|\left\|{\vec{v}}_{i+1}\right\|}$$
(6)$$∆S=S_{i+1}-S_{i}≅\left\|{\vec{v}}_{i}\right\|\left(t_{i+1}-t_{i}\right)$$
where
$t_{i+1}:\mathrm{time}\;\mathrm{at}\;v_{i+1}$$t_{i}:\mathrm{time}\;\mathrm{at}\;v_{i}$
(7)$${\vec{v}}_{i}=\left(v_{i_{x}},\;v_{i_{y}}\right),\;\;\left\|{\vec{v}}_{i}\right\|=\sqrt{{v_{i_{x}}}^{2}+{v_{i_{y}}}^{2}}$$

Then the $k_{m\;i-th}$ curvature and the $r_{i-th}$ radius of curvature at the point $P_{i-th}$ are given by Equations (8) and (9), respectively.
(8)$$k_{m\;i}=\frac{\sqrt{{\left(\left\|{\vec{v}}_{i}\right\|v_{{i+1}_{x}}-\left\|{\vec{v}}_{i+1}\right\|v_{i_{x}}\right)}^{2}+{\left(v_{{i+1}_{y}}\left\|{\vec{v}}_{i}\right\|-\left\|{\vec{v}}_{i+1}\right\|v_{i_{y}}\right)}^{2}}}{\left(t_{i+1}-t_{i}\right){\left\|{\vec{v}}_{i}\right\|}^{2}\left\|{\vec{v}}_{i+1}\right\|}$$
(9)$$r_{i}=\frac{1}{k_{m\;i}}=\frac{\left(t_{i+1}-t_{i}\right){\left\|{\vec{v}}_{i}\right\|}^{2}\left\|{\vec{v}}_{i+1}\right\|}{\sqrt{{\left(\left\|{\vec{v}}_{i}\right\|v_{{i+1}_{x}}-\left\|{\vec{v}}_{i+1}\right\|v_{i_{x}}\right)}^{2}+{\left(v_{{i+1}_{y}}\left\|{\vec{v}}_{i}\right\|-\left\|{\vec{v}}_{i+1}\right\|v_{i_{y}}\right)}^{2}}}$$

And for the $i_{th}$ iteration:(10)$$C_{i}=P_{i}-r_{i+1}r_{i}\left(\frac{{\vec{v}}_{i+1}}{\left\|{\vec{v}}_{i+1}\right\|}-\frac{\vec{v_{i}}}{\left\|{\vec{v}}_{i}\right\|}\right)$$
(11)$$P_{i+1}=R_{i}\left(P_{i}-C_{i}\right)+C_{i}$$
(12)$$R_{i}=\left(\begin{array}{cc} \cos\theta_{i} & \sin\theta_{i}\\ -\sin\theta_{i} & \cos\theta_{i} \end{array}\right)$$
where $R_{i}$ is the two-dimensional rotation matrix and $C_{i}$ is the center of the circle of radius $r_{i}$ (see Figure 5 and Figure 6).

In a circle, the arc length is given as the product of the radius and its respective angle in radians, hence we have $∆S=r_{i}\theta_{i}$
(13)$$\theta_{i}=\frac{\left\|{\vec{v}}_{i}\right\|}{r_{i}}\left(t_{i+1}-t_{i}\right)$$

Since we are dealing with plane curves, it turns out that the curvature has a sign that indicates whether the curve turns to the left or to the right. Such a sign is defined as 1 if the unit tangent vector and the unit normal vector at a point on the curve, both considered to be the base of the plane, coincide with the orientation of the canonical base of the plane, which is the base with unit vectors i and j; otherwise, the value of the sign is −1. One can calculate the sign either by calculating the determinant of the base change matrix, or by calculating the inner product of the unit normal vector with j (of the IMU inner axis). Thus, we introduce $\sigma_{i}$ as the sign of the rotation angle, and it is defined as follows:(14)$$\sigma_{i}=\left\{\begin{array}{c} -1,\;\;\mathrm{if}\;∆\hat{T}\cdot \hat{j}<0\\ 1,\;\mathrm{if}\;∆\hat{T}\cdot \hat{j}>0 \end{array}\right.$$

So that, if the rotation is performed in a clockwise direction, then $\sigma_{i}=-1$; otherwise, when the rotation is performed in counterclockwise direction, we obtain $\sigma_{i}=1$. In the case where $\sigma_{i}=0,$ no rotation is being performed.

For $t_{i}\leq t\leq t_{i+1}$ (See Figure 6)

**Figure 6 micromachines-15-01053-f006:**
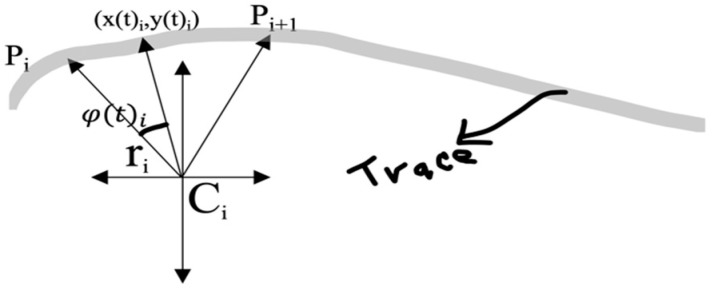
Three successive $P_{i}$ points describing a trace.

The Equation (11) can be rewritten in terms of Cartesian coordinates as Equation (15), where $\varphi_{i}\left(t\right)$ is the angle between two successive points and $(h_{i}$, $k_{i}$) are the Cartesian coordinates of the center of the circle (see Figure 5)
(15)$$\left(\begin{array}{c} x_{i+1}(t)\\ y_{i+1}(t) \end{array}\right)=\left(\begin{array}{cc} \cos\varphi_{i} & \sin\varphi_{i}\\ -\sin\varphi_{i} & \cos\varphi_{i} \end{array}\right)\left(\begin{array}{c} x_{i}-h_{i}\\ y_{i}-k_{i} \end{array}\right)+\left(\begin{array}{c} h_{i}\\ k_{i} \end{array}\right)$$
(16)$$\varphi_{i}(t)=\sigma_{i}\frac{\left\|{\vec{v}}_{i}\right\|}{r_{i}}\left(t_{i+1}-t_{i}\right)$$

Following the construction of letters, it is crucial to recognize that another shape prevails mainly, in addition to the curves; this is the straight line. As mentioned above, efficient construction of the letters generally leads to better legibility and speed, so it is essential to add straight lines to the curves in the process of handwriting generation, since this gives a better projection of the image stored in human memory of what each letter looks like. According to this, the method considering only circular sections can be modified so it can reproduce straight lines, and this modification was called method two.

Method two is based on circular sections and straight lines; assuming a reconstructed trace, knowing its accelerations in the plane and considering the measurement of curvature at point P of a curve (Equation (17)), we have
(17)$$k(t)=\frac{\left\|{\vec{P}}^{\prime }\left(t\right)\times {\vec{P}}^{\prime \prime }\left(t\right)\right\|}{{\left\|{\vec{P}}^{\prime }\left(t\right)\right\|}^{3}}$$
where $\vec{P}\left(t\right)$ is a vector defining a point, given by
(18)$$\vec{P}\left(t\right)=\left(\begin{array}{c} x(t)\\ y(t) \end{array}\right)$$

In addition, the radius of curvature $r\left(t\right)$ of the vector $\vec{P}$ is
(19)$$r(t)=\frac{{\left\|{\vec{P}}^{\prime }(t)\right\|}^{3}}{\left\|{\vec{P}}^{\prime }(t)\times {\vec{P}}^{\prime \prime }(t)\right\|}$$

The known data are the $i_{th}$ accelerations ($a_{i_{x}}\;and\;a_{i_{y}}$) of point P, obtained in the plane of movement of the trace, and the time difference (∆t) between the values obtained for accelerations in the plane. Assuming initial conditions of zero, we have
(20)$$P_{0}=\left[\begin{array}{c} 0\\ 0 \end{array}\right]\;and\;{\vec{v}}_{0}=\left[\begin{array}{c} 0\\ 0 \end{array}\right]$$

Then, if we rewrite Equation (19) in Cartesian coordinates, we obtain
(21)$$r_{i}(t)=\frac{{\left({a_{i_{x}}}^{2}+{a_{i_{y}}}^{2}\right)}^{\frac{3}{2}}({∆t)}^{2}}{\left|a_{i_{x}}a_{{i+1}_{y}}-a_{{i+1}_{x}}a_{i_{y}}\right|}$$

Next, the center of the circle is
(22)$$C_{i}=P_{i}-r_{i}{\hat{n}}_{i}$$
where ${\hat{n}}_{i}$ is the unit normal vector to the curve at $P_{i}$ and is expressed in Cartesian coordinates as
(23)$${\hat{n}}_{i}=\left(\begin{array}{c} -\frac{v_{i_{y}}}{v_{i}}\\ \frac{v_{i_{x}}}{v_{i}} \end{array}\right)$$

The angle of rotation $\theta_{i}$ is
(24)$$\theta_{i}=\arccos\left(\frac{\vec{v_{i}}\cdot {\vec{v}}_{i+1}}{\left\|{\vec{v}}_{i}\right\|\left\|{\vec{v}}_{i+1}\right\|}\right)$$

Then,
(25)$$P_{i+1}=\left\{\begin{array}{c} R_{i}\;\left(P_{i}-C_{i}\right)+C_{i},\;if\;\;\;\left|a_{i_{x}}a_{{i+1}_{y}}-a_{{i+1}_{x}}a_{i_{y}}\right|\neq 0\\ P_{i}+\left\|\vec{v_{i}}\right\|∆t,\;if\;\;\;\left|a_{i_{x}}a_{{i+1}_{y}}-a_{{i+1}_{x}}a_{i_{y}}\right|=0 \end{array}\right.$$

Observe that the condition $a_{i_{x}}a_{{i+1}_{y}}-a_{{i+1}_{x}}a_{i_{y}}\neq 0$ in Equation (25) is sufficient to guarantee the existence of $P_{i+1}$. Otherwise, $r_{i}(t)$ given in Equation (21) is undefined and so is $P_{i+1}$.

## 3. Results

We use patterns of shapes and letters to test the previously established algorithm. In the case of the simulations, these patterns are the points from an equation describing a particular shape. In the second part, we prove the algorithm’s reconstruction of a natural trace using the electronic prototype. The pairs (x,y) to reconstruct the patterns were calculated using Equations (15) and (16) for method one, based only on circle sections; for method two, based on circle sections and straight lines, Equations (24) and (25), are used.

### 3.1. Simulation

The main characteristic of these patterns is that they have a curve morphology. The two primary shapes, curved and straight lines, can describe circles, parabolas, and, more critically, letters (see Figure 7, Figure 8, Figure 9 and Figure 10).

The reference pattern is shown in Figure 7a. The vector time is defined as t=[0, 2π], the *x* coordinated as sin⁡(t), and *y* coordinate as cos⁡(t). It is a circle centered at the origin of the Cartesian coordinate system and drawn from the usual parametric equations. Figure 7b,c show the reconstruction of the reference pattern using method one and method two, respectively. The vector time remains equal; for method one the input was the linear velocity as $\left(\cos\left(t\right),-sin(t)\right)$ and for method two the input was the linear acceleration as $\left(-\sin\left(t\right),-cos(t)\right)$. In these methods, the reference pattern remains unchanged, as we again obtain a circle with a radius of one and a center at the origin. This is not surprising, since both methods construct a curve connecting circular arcs, the osculating circles, with the curve following the curvature vector estimated from the acceleration. The difference between the two methods is that method two considers the case when the curvature is close to zero, assuming a particular error. In such a case, method two constructs a straight-line segment whose extreme points are successive position estimates.

**Figure 7 micromachines-15-01053-f007:**
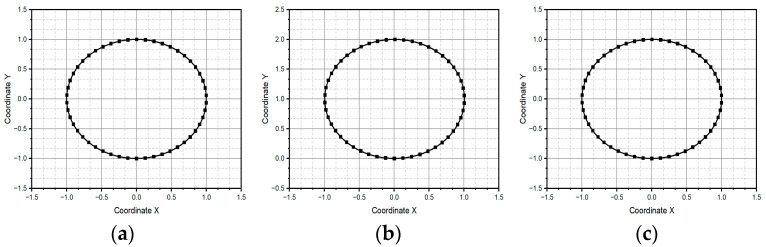
Circular pattern: (**a**) centered on the points (0,0), (**b**) pattern centered on the points (0,1) generated through method one, and (**c**) centered on the points (0,1) generated through method two.

Figure 8a shows a second reference pattern, this time a sinusoid curve; in Figure 8b we observe how method one fails to reconstruct a sinusoid curve as a reference pattern. The main reason behind such a result is that method one does not detect the abrupt curvature changes at the bottom portions of the pattern, i.e., where the curvature slope changes from positive to negative and when it achieves its maximum value. So, method one is unable to capture the quick variations of acceleration. Figure 8c shows how method two successfully reconstructs the reference pattern. It is a consequence of the refinement of adding an exception that constructs straight-line segments when consecutive estimates of acceleration differ in absolute value from those below the considered error in the algorithm of method one. The advantage of straight-line segment construction is that it does not require the knowledge of the curvature sign, i.e., it does not consider when the curve is turning clockwise or counterclockwise. But it fails in reconstructing the smoothness of the curve.

**Figure 8 micromachines-15-01053-f008:**
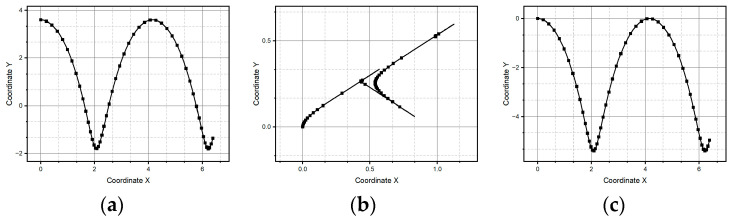
Parabolic pattern: (**a**) ideal case, (**b**) pattern generated through method one, and (**c**) pattern generated through method two.

Figure 9 shows the cursive letter “*a*” as a reference pattern, and it is constructed via the parametric equations of a circle and ellipse. As one can see in Figure 9b, method one fails again to reconstruct the reference pattern because it is unable to properly follow the changes in those points at the curve with abrupt changes in slope, for instance in the small end loop of the letter “*a*”. As evidence of the consistent behavior of method two, Figure 9c shows how such a method recovers the reference pattern acceptably, conserving as much as possible the topology of the trace, despite the different metric features. The pattern of the letter “*a*” is made up of two pairs of equations. The first forms most of the letter, which is the semi-circle, and the second is an ellipse that makes up the residual part of the letter. The connection between both equations forms a straight line, and it is observed that method one fails when trying to complete the letter just when the straight curve begins.

**Figure 9 micromachines-15-01053-f009:**
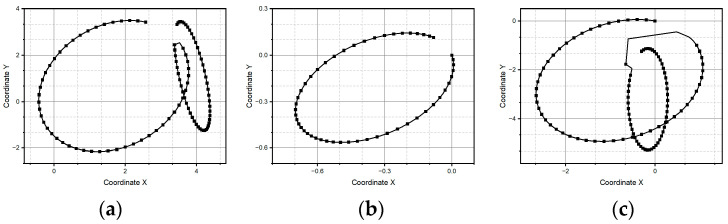
Letter “a” pattern: (**a**) ideal case, (**b**) pattern generated through method one, and (**c**) pattern generated through method two.

As one can see in Figure 10, the reference pattern is the letter e. It is, in a general fashion, an interesting challenge to any same-purpose algorithm, since, despite its intrinsic simplicity, it combines critical features, namely, the topological as well as the metric. It means that there must be, in the reconstructed pattern, an adequate balance between the characteristic upper loop and the bottom ending curve. Both methods reach the goal; the first method produces the letter up to a rotation and a homothety (the reasons for these effects have yet to be studied further). Meanwhile, the second method produces the letter up to just a homothety.

**Figure 10 micromachines-15-01053-f010:**
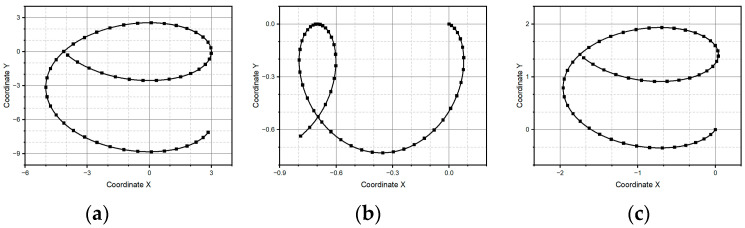
Letter “e” pattern: (**a**) ideal case, (**b**) pattern generated through method one, and (**c**) pattern generated through method two.

### 3.2. Experimental Results

The samples for these experiments are a single trace without placing an additional processing signal, and were obtained using the hardware described in Section 2. The only preprocessing was placing the IMU’s embedded 5 Hz low-pass filter before sending the data to the acquisition and processing board, and a sampling time of 10 ms was used for all the experiments. The experimental tests were carried out by tracing the curve in the air and trying to limit this to an imaginary plane mentally set in the air. The user traced the curve to be reconstructed, seeking to make circular patterns; however, the aim was to make the line as natural as possible without needing a template or forced position control, while not simply leaving the hand free to stroke at a speed that the person feels comfortable with. The test patterns drawn in the air by the person were the same as those used for the simulation.

For method one, only the result for the circular pattern is shown, since it was the only one successfully reconstructed (see Figure 11). Method one has impossibilities for pattern reconstruction in the case of significant curvature variations. In the case of method two, the patterns preserve the metric and the topology; however, it presents homothety. Method two performs better than method one, due to the generation of traverses. The fidelity concerning the original trace depends on the estimation of the initial conditions and how the speed is estimated, which is observed, above all, in the metric of the reconstructed trace. In the case of Figure 11 and Figure 12, despite having a sinusoidal shape in their respective velocities and accelerations (see Figure 13 and Figure 14), completing a cycle of 2π, which in the case of Figure 12 is of higher quality compared to Figure 14, the circle drawn for the circular pattern is successful in both methods. Method one requires calculating linear velocities by setting the radius of gyration between the pen’s tip and the sensor’s center. This is because the IMU measures angular velocities. In method two, the linear accelerations are used directly from the IMU, so it is not necessary to assume conditions that increase the reconstruction’s uncertainty.

After all the previous remarks, we can conclude that method two produces better results in the task of reproducing the critical reference patterns, since it preserves the main topological, as well as metric geometric, features up to rotations, dilations, and translations. Given the above, the remaining test patterns are reconstructed experimentally only using method two, since it successfully reconstructed all patterns (see Figure 13, Figure 14, Figure 15, Figure 16, Figure 17, Figure 18, Figure 19 and Figure 20). For the parabolic pattern of Figure 15, the acceleration in the *y*-axis (see Figure 16) has a baseline; this is because, in the case of the morphology of this pattern, the displacement on this same axis has the same baseline. The letter a (see Figure 17) has a reconstruction based on the measured accelerations in Figure 18, while the letter e (see Figure 19) is reconstructed from the acceleration patterns measured in Figure 20 for both cases, maintaining the base circularity of the letter and allowing the identification of the mark with the naked eye.

## 4. Discussion

The main objective of this work is to demonstrate that it is possible to generate a non-inertial frame online cursive-handwriting system from curvature models, since this is the predominant form in this type of writing.

In the comparison made between the two proposed methods, we can highlight the fact that both show that it is possible to construct letters from linking multiple circular sections and that the patterns obtained are very similar to what is recognized. However, the legibility between one and the other is far from equal, since our understanding is much closer to forming letters when they are made up of curves and straight lines. Even in cursive writing, after all, the notes that only include straight lines or simple curves are the first that the human mind can identify.

That said, the method that uses circular sections and straight lines is the most complete in reconstructing cursive letters and for the generation of online handwriting with a non-inertial frame of reference. This work obtains an object focused on feature engineering based on differential geometry, prioritizing the shapes that the patterns present and not the dimension or accuracy of the position of the points, since it is aiming to be used in machine learning algorithms where it is necessary to vectorize the normalized feature.

Making a qualitative visual comparison with the results previously reported in the literature [5,8], it is possible to affirm that the results obtained with method two are comparable in terms of the fidelity of tracking the fundamental shape of the stroke. It is worth noting that, in contrast to the methods reported in [5] and [8], the method proposed in this work does not rely on an exhaustive preprocessing of the data in order to obtain the reported results.

The movement of the pen could be performed in three dimensions. However, this work assumes that the strokes made by the person are limited to an imaginary plane mentally set to write in the air. Both methods can be expanded to the idea of tracing in three dimensions; however, reading three-dimensional traces could be unpractical.

## 5. Conclusions

The methods shown are an alternative to the use of orientation- and trajectory-tracing tools for several reasons. The first reason is that they do not require a specific work area with special conditions such as a fixed work surface, or environmental conditions such as fixed lighting.

Comparing the two proposed models shows that it is possible to construct letters from circular sections and that the patterns obtained are similar to what is recognized, providing an alternative to quaternion calculus for poorly specified orientation problems.

The first method, based only on oscillating circles, has the advantage of a lower computational cost in the calculation. However, it has the disadvantage of not being sensitive to abrupt changes in trajectories. For method two, a combination of circles and straight lines is used; it was possible to observe that the recreation of the trajectory is faithful to the original trace, which make this method more suitable for an online handwriting system without an inertial reference frame and, even more, it could be extrapolated to other applications such as leader–follower robotic systems or teleoperation applications.

## Figures and Tables

**Figure 1 micromachines-15-01053-f001:**
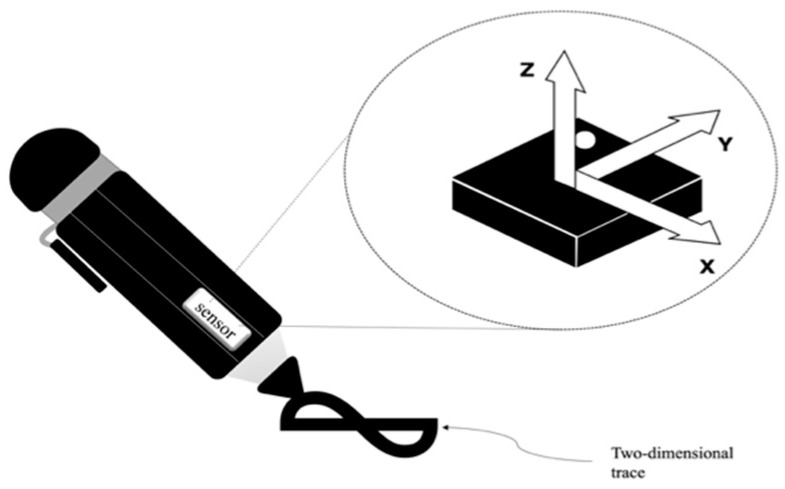
Digital writing system, a common pen with an IMU.

**Figure 2 micromachines-15-01053-f002:**
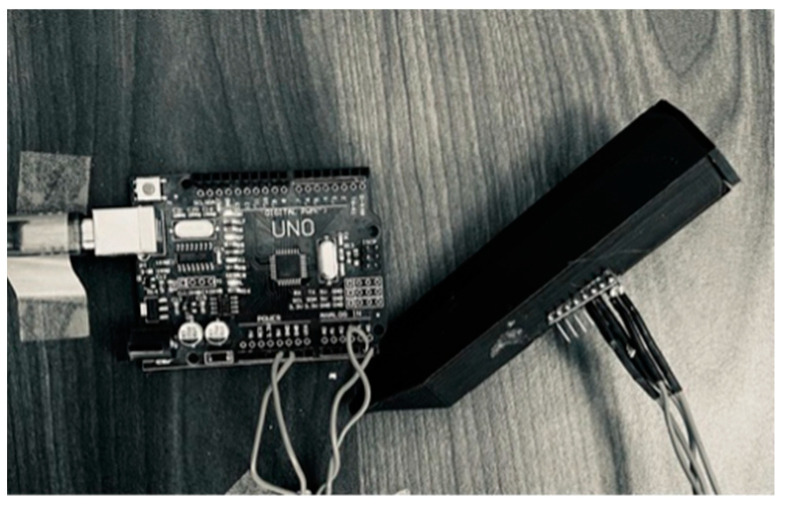
Physical digital writing system, connected to an UNO Arduino board.

**Figure 3 micromachines-15-01053-f003:**
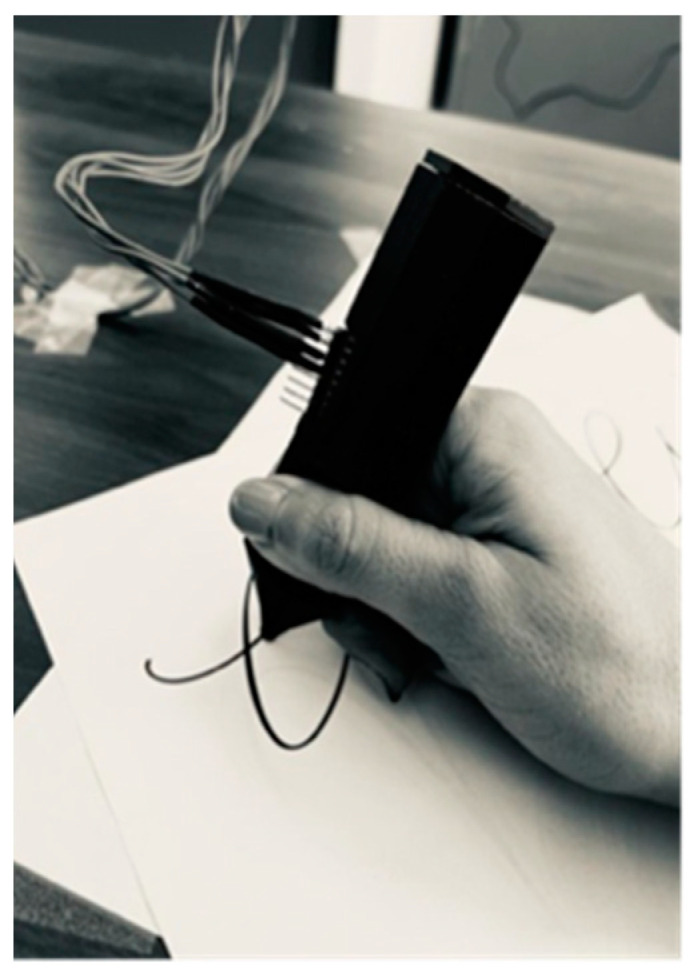
Friendly prototype with the shape of a pen.

**Figure 4 micromachines-15-01053-f004:**
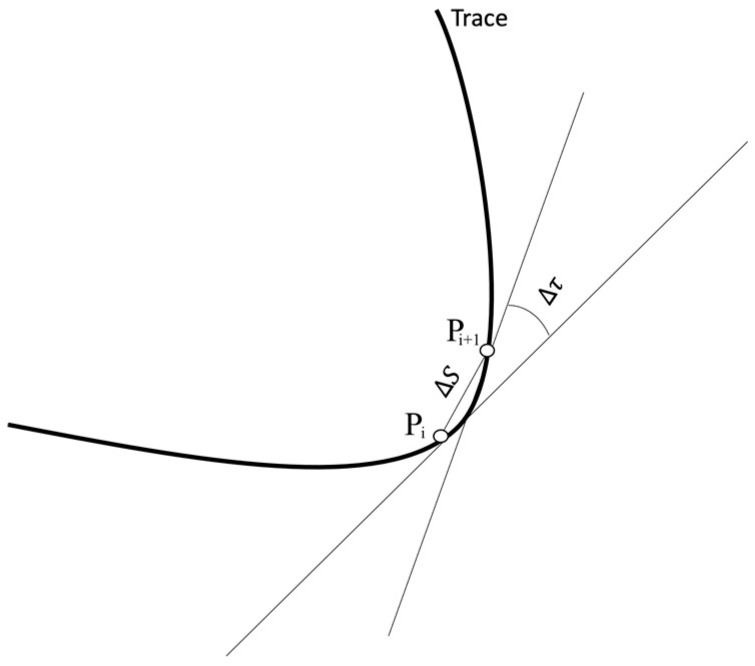
Let $P_{i+1}$ be a second point over the curve close to point $P_{i}$. The point of contact to the tangent describes an arc ∆S, and the tangent rotates through an angle ∆τ, known as the contingency angle.

**Figure 5 micromachines-15-01053-f005:**
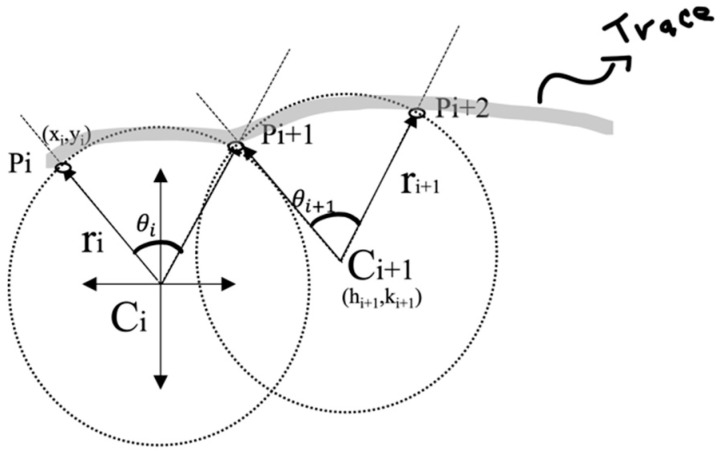
Three successive P points describing a trace with the circles associated with it.

**Figure 11 micromachines-15-01053-f011:**
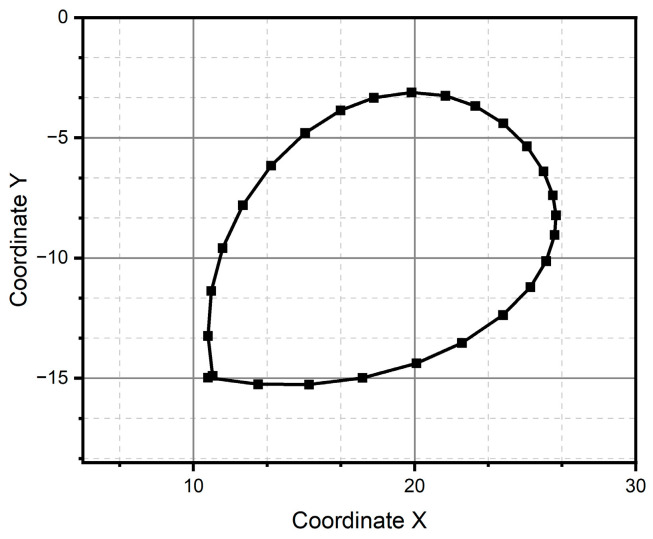
Pattern of a circle generated through method one.

**Figure 12 micromachines-15-01053-f012:**
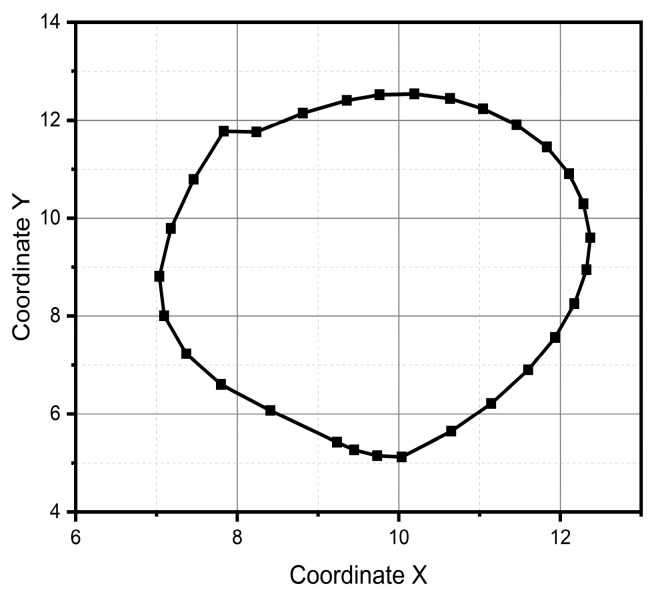
Pattern of a circle generated through method two.

**Figure 13 micromachines-15-01053-f013:**
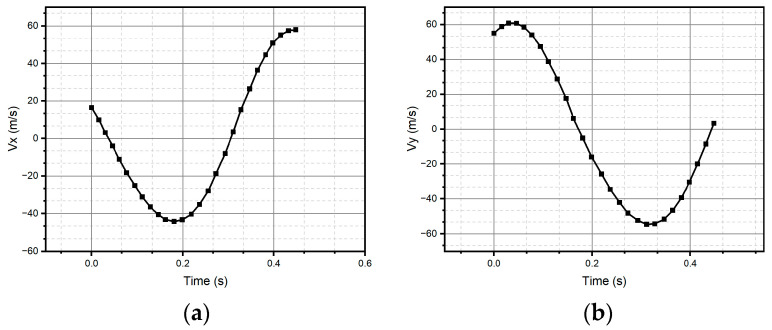
Velocities (**a**) $V_{x}$ and (**b**) $V_{y}$ were measured through the IMU and used to reconstruct the circle.

**Figure 14 micromachines-15-01053-f014:**
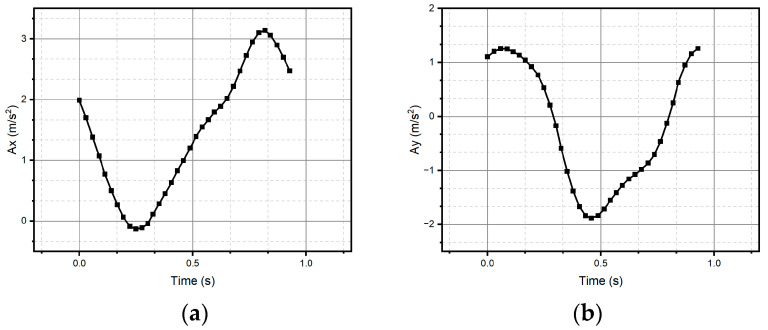
Accelerations (**a**) $A_{x}$ and (**b**) $A_{y}$ were measured through the IMU and used to reconstruct the circle.

**Figure 15 micromachines-15-01053-f015:**
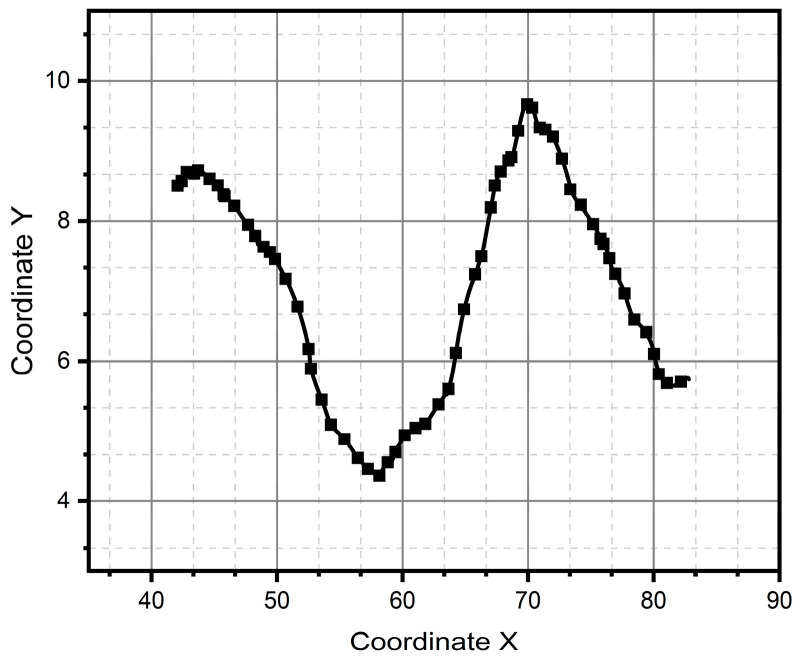
Parabolic pattern generated through method two.

**Figure 16 micromachines-15-01053-f016:**
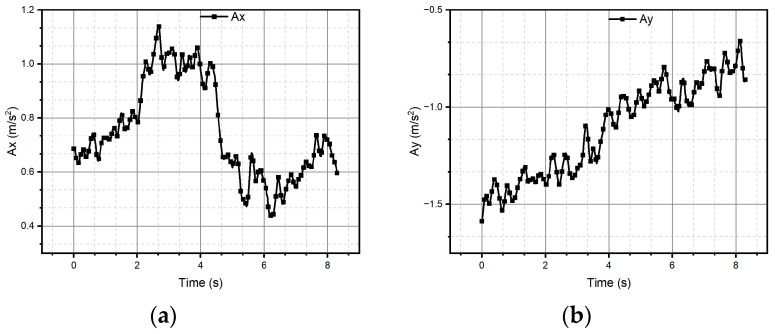
Accelerations (**a**) $A_{x}$ and (**b**) $A_{y}$ were measured through the IMU and used to reconstruct the parabolic pattern.

**Figure 17 micromachines-15-01053-f017:**
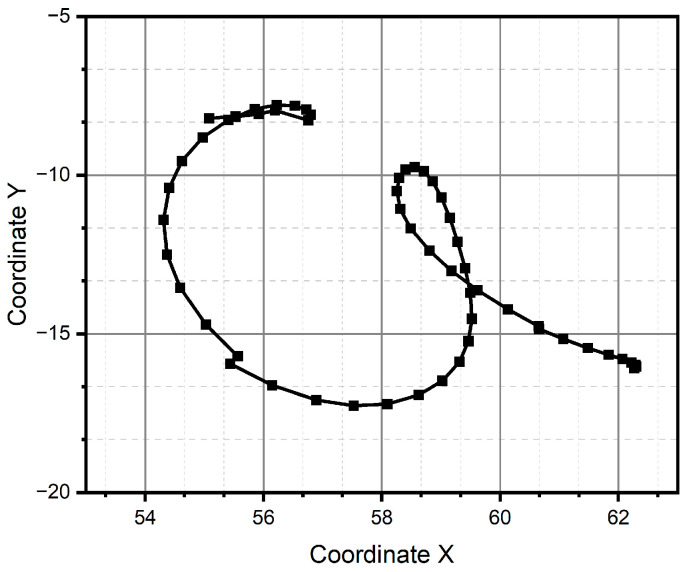
Reconstructed trace of the handwriting letter “a” generated through method two.

**Figure 18 micromachines-15-01053-f018:**
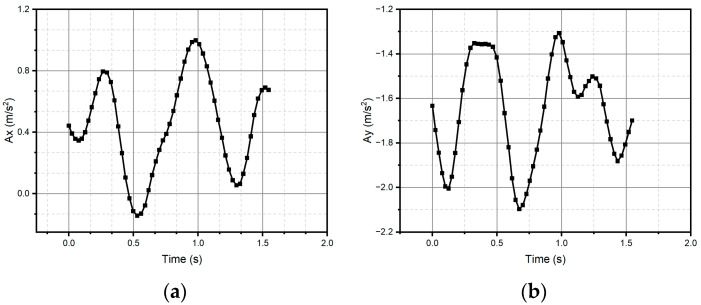
Accelerations (**a**) $A_{x}$ and (**b**) $A_{y}$ were measured through the IMU and used to reconstruct the letter “a”.

**Figure 19 micromachines-15-01053-f019:**
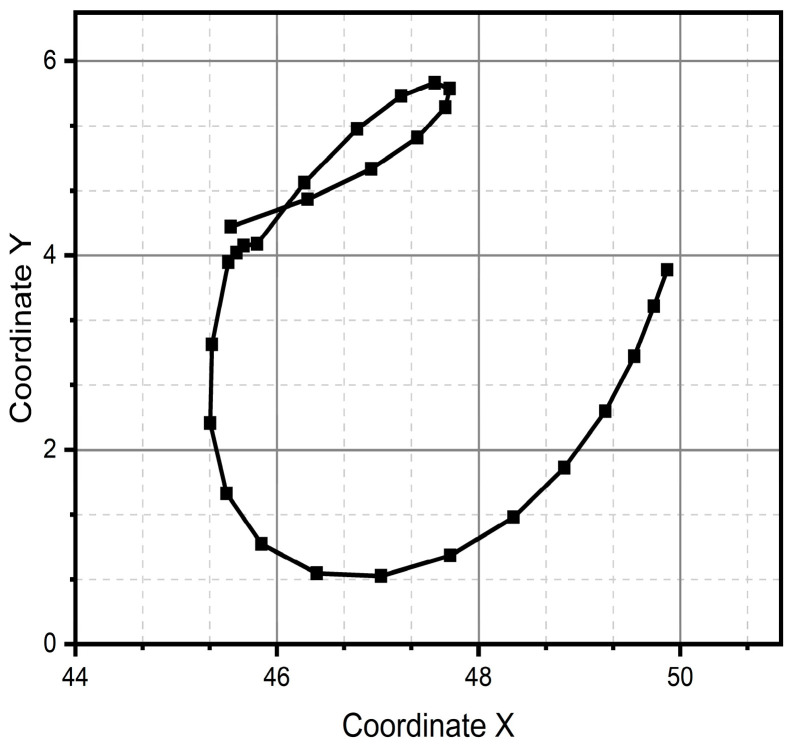
Reconstructed trace of the handwriting letter “e” through method two.

**Figure 20 micromachines-15-01053-f020:**
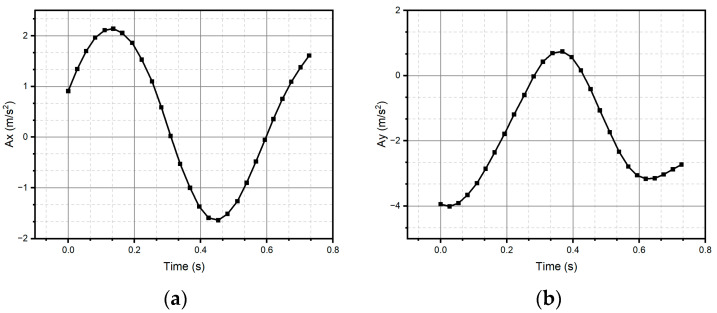
Accelerations (**a**) $A_{x}$ and (**b**) $A_{y}$ were measured through the IMU and used to reconstruct the letter “e”.

## Data Availability

Data supporting reported results can be found here.

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
