# Peer review of "Online Handwriting Recognition Method with a Non-Inertial Reference Frame Based on the Measurement of Linear Accelerations and Differential Geometry: An Alternative to Quaternions"

_micromachines, 2024, doi:10.3390/mi15081053_

Round 1

Reviewer 1 Report

Comments and Suggestions for Authors

The authors present an alternative method to use orientation trajectory tracing tools with more relaxed conditions to obtain fine writing resulsts when using a pen with an IMU.

Minor issues detected:

- In equation 4. Delta T is a vector, and in equation 5, Delta T is a scalar value. In equation 5 the formula is the modulus of Delta T, which is consistent with equation 2.

- Provide a citation for equation 7.

- In method 1 (eq. 9) the authors use the velocity and for the method 2, the authors use the acceleration (equation 21), please provide a clarification why use different inputs in both methods.

Author Response

Thank you so much for the time invested in the review of this paper, regarding your comments our responses are the following:

COMMENTS 1: In equation 4. Delta T is a vector, and in equation 5, Delta T is a scalar value. In equation 5 the formula is the modulus of Delta T, which is consistent with equation 2.

RESPONSE 1: Thank you for pointing this out, there was a mistake in the notation used in the equations which has already been corrected.

COMMENTS 2: Provide a citation for equation 7.

RESPONSE 2: Equation 7 also presented an inconsistent notation, that has already been corrected. Given the correction made we believe there’s no need for a citation.

COMMENTS 3: In method 1 (eq. 9) the authors use the velocity and for the method 2, the authors use the acceleration (equation 21), please provide a clarification why use different inputs in both methods.

RESPONSE 3: Thank you for the comment, effectively there wasn’t a clear explanation of why there are different inputs for both methods, we modify the explanations given in the paper to make clearer (line 153 to 161). In method one we use the angular velocity direct as measured by the IMU as the input data and we transform it to linear velocities assuming the tip of the pen as the center of the rotation of the IMU, these method permits a short computation time. On the other hand, for method two we consider the linear acceleration measured by the IMU as input in order to eliminate the transformation made to the angular velocity and also to be able to add the straight-line recognition, not only the circular sections as in method 1.

Once again, we appreciate your time and attention in reviewing this work and we hope that your comments have been addressed accurately.

Greetings

Reviewer 2 Report

Comments and Suggestions for Authors

The paper develops a method of reconstruction of handwriting trajectory using the velocity/acceleration histories. Proposed method is of considerable interest, but presentation requires substantial improvement.

1. Introduction must be improved. At least first two paragraphs needs rewriting. There is a lot of general discussion and too few words about the state of the art in the handwriting recognition. The Authors refer to works, but not discuss them; for instance: “There are various efforts to improve the presentation of this handwriting online to reproduce faithfully the strokes of the pen movement [1-8]”, “Quaternions are a popular and efficient alternative to solve this problem and to creating orientation systems using IMUs [16-25]”.

2. In general, the Introduction does not provide clear picture of the background in the field and novelty of this work? The proposed approach is relatively simple, is it really original in the field of writing recognition?

3. Please, clearly designate vector values through the text and use different notations for scalar and vector values. For instance, “\Delta\hat T” is a vector in Eq. (4) and the same notation is a scalar in Eq. (5). This notation system is messleading.

4. Line 158: “𝐶i is the center of the circle of radius 𝑃i𝑟i”? What does it mean radius as a difference between vector Pi and scalar ri?

5. Eq. (14): What if the product is zero, it means “\Delta\hat T” is perpendicular to “j”?

6. In Fig. 4 and Eq. (1), P’ means the next point, while in Eq. (17) it is the first derivative with respect to time.

7. Section 3.1. As this work is methodological, the restored patters are not enough. It is recommended to describe, at least by one example, how the reference patter is transformed to accelerations, and how these accelerations are used to restore curvatures, circle center etc. I mean, to describe with corresponding plots.

8. The main difference in two methods is recognition of straight lines in the second one. What is the reason of pure quality of the first method for the patterns without straight lines as in Figs. 9 and 10? It seems that an error in algorithm is more probable reason.

9. Another difference between two methods is using of velocities for the first one and accelerations for the second one. I can suppose that the reason of poor quality of the first method is unequal quality of the velocity and acceleration histories used for trace reconstruction.

10. How did you choose the time step between points and how sensitive are the results to this time step?

11. How did you transform the 3D pattern of velocity/acceleration into 2D data? In other words, how did you choose the direction perpendicular to writing plane basing on accelerometer signals?

12. Discussion section does not provide comparison with other methods and results in the field, which makes the significance of presented results limited.

13. Typos. Line 153: “radio(?)”. Line 183: “que”?

14. English must be refined. There are a lot of unclear sentences in the text requiring reformulation.

Comments on the Quality of English Language

English must be refined. There are a lot of unclear sentences in the text requiring reformulation.

Author Response

Thank you so much for the time invested in the review of this paper, regarding your comments our responses are the following:

COMMENTS 1: Introduction must be improved. At least first two paragraphs needs rewriting. There is a lot of general discussion and too few words about the state of the art in the handwriting recognition. The Authors refer to works, but not discuss them; for instance: “There are various efforts to improve the presentation of this handwriting online to reproduce faithfully the strokes of the pen movement [1-8]”, “Quaternions are a popular and efficient alternative to solve this problem and to creating orientation systems using IMUs [16-25]”.

RESPONSE 1: Thank you for the comment, the introduction has been rewritten taking into account this observation.

COMMENTS 2: In general, the Introduction does not provide clear picture of the background in the field and novelty of this work? The proposed approach is relatively simple, is it really original in the field of writing recognition?

RESPONSE 2: Thank you for the comment, the introduction has been rewritten taking into account this observation.

COMMENTS 3: Please, clearly designate vector values through the text and use different notations for scalar and vector values. For instance, “\Delta\hat T” is a vector in Eq. (4) and the same notation is a scalar in Eq. (5). This notation system is messleading.

RESPONSE 3: Thank you for pointing out this, there were inconsistencies in the notation through the equations that have already been corrected. The differences between the notation of scalars and vectors were unified in the full text.

COMMENTS 4: Line 158: “?i is the center of the circle of radius ?i − ?i”? What does it mean radius as a difference between vector Pi and scalar ri?

RESPONSE 4: Again, this was an inconsistency that has already been corrected. The circle's radius is only ?i.

COMMENTS 5: Eq. (14): What if the product is zero, it means “\Delta\hat T” is perpendicular to “j”?

RESPONSE 5: In the case when 〖 σ〗_i=0 no rotation is being performed. This was already clarified in the text.

COMMENTS 6:  In Fig. 4 and Eq. (1), P’ means the next point, while in Eq. (17) it is the first derivative with respect to time.

RESPONSE 6: Figure 4 has now a different notation to avoid this confusion.

COMMENTS 7:  Section 3.1. As this work is methodological, the restored patters are not enough. It is recommended to describe, at least by one example, how the reference patter is transformed to accelerations, and how these accelerations are used to restore curvatures, circle center etc. I mean, to describe with corresponding plots.

RESPONSE 7: In attention to this comment the vectors of one of the reference patterns were described in detail: the time vector, the x and y coordinate vectors, the velocity vectors for method one, and the acceleration vectors for method two.

COMMENTS 8: The main difference in two methods is recognition of straight lines in the second one. What is the reason of pure quality of the first method for the patterns without straight lines as in Figs. 9 and 10? It seems that an error in algorithm is more probable reason. The failure of method one is explained more broadly in the case of the pattern in Figures 9 and 10

RESPONSE 8: We appreciate your comment on this, implementation of the method one for letters a and e was reviewed and run once again, obtaining the same results, so we discard and error in the algorithm as the source of the error. Then for letter “a” the poor quality of the results is attached to its incapability to respond to abrupt changes in curvature and non-recognition of straight lines, these results were the reason why we proposed method two to overcome these errors. Regarding the errors in letter “e”, we are still working in the source of the rotation and scale effects, as these are not clearly identified yet (a note mentioning this was added in line 328).

COMMENTS 9: Another difference between two methods is using of velocities for the first one and accelerations for the second one. I can suppose that the reason of poor quality of the first method is unequal quality of the velocity and acceleration histories used for trace reconstruction.

RESPONSE 9: Based on the comment, the results section was expanded with the comparison between methods one and two and the possible sources of failure. We also want to highlight that both velocity and acceleration histories are obtained with the same filtering and sampling parameters, so we discard a lesser quality velocity history as a source of error, perhaps the transformation from angular to linear velocities is a more likely source of error.

COMMENTS 10:  How did you choose the time step between points and how sensitive are the results to this time step?

RESPONSE 10: Thank you for pointing out this, as it wasn’t mentioned in sufficient clarity in the text, a word on this was added in section 3, line 347.

COMMENTS 11:  How did you transform the 3D pattern of velocity/acceleration into 2D data? In other words, how did you choose the direction perpendicular to writing plane basing on accelerometer signals?

RESPONSE 11: Again, this was not clearly explained in the text, thanks for pointing that out. Line 116 details this comment and explains how two-dimensional vectors are used.

COMMENTS 12: Discussion section does not provide comparison with other methods and results in the field, which makes the significance of presented results limited.

RESPONSE 12: In the discussion a paragraph was added referring to the comparison with other methods. It is worth noting that the comparison was only qualitative.

COMMENTS 13: Typos. Line 153: “radio(?)”. Line 183: “que”?

RESPONSE 13: Typos were corrected

COMMENTS 14: English must be refined. There are a lot of unclear sentences in the text requiring reformulation.

RESPONSE 14: Agree. We have accordingly reviewed the full text and rewritten when necessary.

Once again, we appreciate your time and attention in reviewing this work and we hope that your comments have been addressed accurately.

Greetings

Round 2

Reviewer 2 Report

Comments and Suggestions for Authors

My comments have been sufficiently addressed.